# Recent Advancements in Antifibrotic Therapies for Regression of Liver Fibrosis

**DOI:** 10.3390/cells11091500

**Published:** 2022-04-29

**Authors:** Anshika Jangra, Ashish Kothari, Phulen Sarma, Bikash Medhi, Balram Ji Omar, Karanvir Kaushal

**Affiliations:** 1Department of Research and Development, Cure Therapeutics Inc., Suwon 16506, Korea; anshika.jangra@curetherapeutics.com; 2Department of Microbiology, All India Institute of Medical Sciences, Rishikesh 249203, India; ashishkothari1212@gmail.com (A.K.); balram.micro@aiimsrishikesh.edu.in (B.J.O.); 3Department of Pharmacology, Post Graduate Institute of Medical Education and Research, Chandigarh 160012, India; phulen10@gmail.com (P.S.); medhi.bikash@pgimer.edu.in (B.M.); 4Department of Biochemistry, All India Institute of Medical Sciences, Rishikesh 249203, India

**Keywords:** hepatic stellate cells, urokinase plasminogen activator receptor, miRNA, tumor necrosis factor-stimulated gene, mesenchymal stromal cells, natural killer cells

## Abstract

Cirrhosis is a severe form of liver fibrosis that results in the irreversible replacement of liver tissue with scar tissue in the liver. Environmental toxicity, infections, metabolic causes, or other genetic factors including autoimmune hepatitis can lead to chronic liver injury and can result in inflammation and fibrosis. This activates myofibroblasts to secrete ECM proteins, resulting in the formation of fibrous scars on the liver. Fibrosis regression is possible through the removal of pathophysiological causes as well as the elimination of activated myofibroblasts, resulting in the reabsorption of the scar tissue. To date, a wide range of antifibrotic therapies has been tried and tested, with varying degrees of success. These therapies include the use of growth factors, cytokines, miRNAs, monoclonal antibodies, stem-cell-based approaches, and other approaches that target the ECM. The positive results of preclinical and clinical studies raise the prospect of a viable alternative to liver transplantation in the near future. The present review provides a synopsis of recent antifibrotic treatment modalities for the treatment of liver cirrhosis, as well as a brief summary of clinical trials that have been conducted to date.

## 1. Background

Chronic liver diseases are becoming a global health burden; they are responsible for approximately 2 million deaths per year, with approximately 1 million deaths resulting from complications associated with cirrhosis [1]. The mortality rate varies significantly across different geographical regions. In places such as Central Asia, Central Europe, Eastern Europe, and Central America, it is regarded as a public health priority. Furthermore, only about one-third of countries have accurate mortality data [1]. Increased mortality may be linked to viral hepatitis, but other factors—such as alcoholic and non-alcoholic fatty liver diseases, which are currently the most common in developed countries—may also be contributing factors [2]. Because of the increasing morbidity and mortality associated with chronic liver diseases, it is imperative that action be taken immediately to prioritize the identification and treatment of patients [3].

Over the last decade, the number of studies on organ fibrosis has increased, and common characteristics of fibrosis across different tissues have been identified. While tissue fibroblasts are considered to be heterogeneous, myofibroblasts exhibit similar phenotypes and molecular characteristics in fibrotic organs such as the lungs, kidneys, and liver, indicating that a conserved pathogenic pathway is common in fibrotic organs of different origins [4]. Initially, the development of advanced fibrosis was thought to be associated with specific signaling pathways. However, the signaling networks in fibrotic diseases have been overlooked, and a comprehensive understanding of molecular pathways is required to understand the disease. Because of the activation of the myofibroblast cell type, the pathogenesis of fibrosis shares its etiology with altered epithelial–mesenchymal cell interactions, inflammation, and fibroblast proliferation [5]. The TGFβ-SMAD2/3 signaling pathway, which is involved in cell transdifferentiation into myofibroblasts and abnormal extracellular matrix deposition, is another common pathway [6].

The progression of fibrosis into cirrhosis is determined by the development of portal hypertension, the activation of systemic inflammation, and the development of hyperdynamic circulatory dysfunction [7]. Portal hypertension is a detrimental complication that can obstruct portal blood flow [8]. In the case of cirrhosis, increased intrahepatic vascular resistance to the portal flow can elevate portal pressure. In patients with chronic liver diseases, portal hypertension can be driven by progressive fibrosis and intrahepatic vasoconstriction [9,10]. Because these diseases do not predictably manifest themselves, they are associated with varying mortality risks, as previously stated. As a result, a multistate approach to describing the clinical course has been considered. This necessitates the evaluation of the probabilities of various outcomes. This review aims to summarize relevant clinical states, possible regression methodologies, and design frameworks that may apply to multistate models of fibrosis.

### 1.1. Hepatic Scarring and Extracellular Matrix Leading to Fibrosis

Fibrogenesis is regarded as a normal wound-healing mechanism that occurs in response to any type of tissue damage. The activation of fibrogenic pathways by hepatocellular injuries results in the secretion of fibrogenic components into the extracellular matrix which, in turn, results in the formation of scar tissue [11,12]. In a normal liver, the ECM regulates a balance between synthesis and degradation [13]. Regulated amounts of glycoproteins—such as fibronectin and laminin—type IV collagen (non-fibrogenic), and proteoglycans, such as heparan sulfate, are also found in the collagen fibers [14,15]. However, during chronic injury, the ECM production exceeds that of its degradation by 6–8-fold, leading to the thickening of fibrotic septa and the crosslinking of collagen [12]. As illustrated in Figure 1, the non-fibrogenic type IV collagen is gradually replaced by the fibrogenic type I and II collagen [12].

The involvement of parenchymal and non-parenchymal liver cells, as well as infiltrating immune cells, complicates the process of the progression and resolution of fibrosis [16,17]. Additionally, smooth-muscle actin, fibronectin, and hyaluronic acid are secreted into the ECM [18]. Cirrhosis develops as a result of disease progression and sustained fibrogenesis. It is defined by the end-stage accumulation of scarring as well as the distortion of liver cells and their vasculature [19]. While the response to tissue injury is rapid, persistent or repeated injury results in the death of hepatocytes via apoptosis and necrosis, which is a critical step in activating inflammatory and pro-fibrogenic pathways, thereby initiating the progression of fibrosis [12,16].

Most chronic liver diseases, such as viral hepatitis B and C or alcoholic (ASH) and non-alcoholic steatohepatitis (NASH), can progress to the formation of fibrotic tissue and scarring [20,21,22]. Persistent liver injury can activate HSCs, which are one of the primary sources of hepatic scarring [23]. HSCs are classified into four distinct phenotypes—quiescent, activated, inactivated, and senescent—each of which plays a critical role in liver fibrosis [24]; their phenotypes are highlighted in Table 1. Activated HSCs can undergo apoptosis or revert to an inactivated phenotype that is similar to but distinct from quiescent HSCs during regression [25]. Immune cells can aid in fibrogenesis and fibrosis regression by facilitating the reabsorption of the fibrous scar [26]. 

KCs, NK cells, NKT cells, and dendritic cells coordinate with antigen-presenting cells—T and B cells—to connect the innate and adaptive immunity in the liver. An imbalanced inflammatory milieu following persistent injury to the liver gradually advances to liver fibrosis [30]. Recently, various novel immune cells—including T-helper 17 (Th17), regulatory T cells (Tregs), mucosa-associated invariant T cells (MAITs), and their related cytokines have been reported to regulate liver fibrosis [31,32,33]. Th17 cells—a subset of CD4+ T show fibrogenic properties owing to their interleukin (IL)-17, IL-22, and IL-23 production [34,35]. High levels of intrahepatic Th17 and IL-17 are commonly observed in liver fibrosis caused by various etiologies [36,37]. IL-17A promotes the transformation of HSCs into myofibroblasts and the production of collagen through the STAT3 signaling pathway [35]. The Tregs/Th17 ratio is found to be altered during liver fibrosis progression. The JAK inhibitor “tofacitinib” has been shown to restore the Tregs/Th17 balance, thereby alleviating liver fibrosis [38,39]. The MAIT cells account for approximately 30% of all CD3+ T cells present in the liver, and can be stimulated by IL-12 and IL-18 to secrete IFN-γ and granzyme [40,41]. MAIT cells are also found to promote the production of pro-inflammatory cytokines, such as IL-6 and IL-8, in mono-derived macrophages in animal models [42]. High levels of CXCR6+ T cells (auto-aggressive) have been detected in the blood of hepatitis-C–infected patients compared with healthy controls, and > 60% of intrahepatic human T cells expressed CXCR6, including CD4, CD8, and CD56+ (NK) T cells [43]. CXCL16 is a ligand for CXCR6+, and has been detected in the hepatocytes and bile ducts of patients with liver disease [43], as well as in murine liver sinusoidal endothelial cells [44] and in the macrophages or dendritic cells of other organs. Functionally, CXCR6+ CD8 T cells express granzyme, TNF-α, IFN-γ, and programmed cell death protein 1 (PD-1), suggesting an activated exhausted phenotype during NASH. In a sequential process triggered by IL-15, metabolic signals such as acetate and extracellular ATP activate CXCR6+ CD8 [45] T cells that promote non-specific killing of hepatocytes and instigate disease progression [45]. 

Single-cell RNA sequencing (scRNA-Seq) has uncovered the complex cell-to-cell interactions of non-parenchymal cells in chronic liver diseases and cirrhosis. scRNA-Seq analysis of healthy and cirrhotic human livers unveiled heterogeneity in fibrosis-associated non-parenchymal cells [46]. Ramachandran et al. showed the presence of a specific macrophage subpopulation—i.e., scar-associated macrophages (SAMs)—to be more prevalent in cirrhotic tissue in comparison to the healthy liver. These SAMs can activate HSCs, and promote mesenchymal cell activation and fibrogenesis. scRNA-Seq analysis also unveiled the role of platelet-derived growth factor receptor-α (PDGFRα) mesenchymal cells in scar-associated mesenchymal cells and two previously unknown subpopulations of scar-associated endothelial cells [46]. Understanding the molecular mechanism underlying hepatic fibrogenesis is critical for developing novel therapeutic strategies [25].

### 1.2. Fibrosis Reversibility and Possible Regression

Recent evidence refutes the long-held belief that cirrhosis (a more advanced form of fibrosis) is always irreversible [47,48]. In certain animal experiments, removing the causative agent resulted in cirrhosis regression [49]. For instance, in the carbon tetrachloride (CCl4) and bile duct ligation (BDL) models of cirrhosis, cessation of the injury results in resolution of the cirrhosis [47]. Regardless of the initial cause of injury, fibrosis marks a common pathway for chronic hepatic inflammation. However, since the wound-healing response is a dynamic process, it has the potential to resolve without scarring. The removal of the injurious stimulus has been shown to improve fibrosis in viral hepatitis [50,51], alcoholic liver disease [52], biliary obstruction [50], and autoimmune hepatitis [51]. Thus, the aspect common to all cases of cirrhosis improvement is the eradication of the underlying causative agent. Effective fibrolysis requires the targeting of several mechanisms, including (a) ECM degradation, (b) myofibroblast deactivation, (c) hepatocyte regeneration, and (d) vascular and parenchymal remodeling.

Although the precise point at which cirrhosis becomes irreversible is unknown, chronic damage results in an increase in the production of acellular and thick fibrotic septa that are resistant to degradation [53]. Collagen—the most abundant ECM scaffolding protein, is crosslinked and wrapped in elastin filaments, which results in decreased matrix metalloprotein (MMP) expression/activation [54]. Additionally, ECM crosslinking affects myofibroblast behavior, and an insoluble ECM prevents myofibroblasts from deactivating [18]. The efficiency of fibrolysis is determined by several factors—most notably the HSC population and is modulated by the ability of hepatocytes to regenerate and the response of inflammatory cells to repeated injury. The immune system plays a critical role in the progression and regression of fibrosis, and macrophages play a critical role in wound healing [26]. Apart from matrix degradation, fibrosis resolution requires the deactivation/eradication of activated HSCs.

The role of platelets in improving liver fibrosis has been proven. Studies have shown that platelets can decrease collagen production by inactivating HSCs [55]. Growth factors such as platelet-derived growth factor (PDGF) and hepatocyte growth factor (HGF) promote liver regeneration [55]. The regenerative effect of platelets in the liver involves a direct effect on hepatocytes, a cooperative effect with liver sinusoidal endothelial cells, and a collaborative effect with KCs [55].

### 1.3. Potential Targets for Cellular Therapy

Antifibrotic therapy primarily targets activated HSCs, (portal) myofibroblasts, and the ECM. However, HSCs and myofibroblasts have the ability to communicate with a wide variety of cell types, and can initiate fibrogenesis, induce their quiescence and apoptosis, or even remove excess ECM via the release of fibrotic enzymes and phagocytosis [16]. These pathways may collectively contribute to the development of additional or complementary pharmacological targets. Direct-action antifibrotics are those that target HSCs, myofibroblasts, and the ECM, whereas indirect-action antifibrotics target other cell types and their pathways [16].

When developing antifibrotic treatment strategies, it may be beneficial to consider two major multicellular functional units that can contribute to fibrosis, depending on the etiology and stage of liver disease: (1) perisinusoidal/vascular—consisting of pericytes such as HSCs, sinusoidal endothelial cells, macrophages/KCs, NKs, and other inflammatory cells, as well as hepatocytes; (2) portal/periportal—consisting of cholangiocytes/ductular cells, portal fibroblasts, and myofibroblasts, as well as various inflammatory cells such as T and B cells and macrophages/dendritic cells [16,56,57]. Other cellular interactions occurring within these units include various growth factors, cytokines, and proteases that are potentially useful in the treatment of fibrosis. During fibrosis, the cellular interaction within and between these multicellular units is distorted as a result of the chronic wound-healing response, which is characterized by the deposition of excessive scar tissue and the remodeling of the blood vessels [58]. As previously stated, even in patients with advanced fibrosis/cirrhosis, when the injury is eliminated, there can be a reversal in liver architecture, as seen in patients treated for chronic hepatitis B and C infections [59,60]. Although there have been reversals, it is a slow process—particularly for patients with advanced fibrosis. As a result, there has been little progress in the development of antifibrotic therapies that are both effective and free of major side effects [16]. Furthermore, these therapies should carefully modulate multicellular units toward fibrolysis while simultaneously directing them toward non-fibrotic tissue maintenance. As a result, combination therapies that target two or more key cellular molecules or units will be required.

## 2. Designing the Framework for Developing Antifibrotic Therapies

### 2.1. Role of Non-Invasive Biomarkers

In recent years, there has been an increase in the need for and interest in identifying liver fibrosis through the use of non-invasive surrogate markers [61]. Both clinicians and patients may find serum markers of liver fibrosis to be an appealing and cost-effective alternative to liver biopsy in certain situations [56]. The non-invasive markers of liver fibrosis are split into two approaches: one is a physical approximation to measure liver stiffness, and the other used blood-based biomarkers. The stiffness and decreased elasticity of fibrotic liver can be assessed using transient elastography (TE) and magnetic resonance elastography (MRE), along with many other radiological techniques. Serum-based liver fibrosis markers are extensively evaluated and are typically divided into direct and indirect markers. Direct markers of fibrosis are smaller or larger fragments of the ECM released into the circulation during ECM turnover. They are further classified into matrix deposition (fibrogenesis-linked) and matrix degradation (fibrolysis-linked) markers. Indirect markers are routine laboratory tests reflecting hepatic alteration. Some algorithms use radiological, direct, or indirect panels of tests in different combinations. These combinational biomarkers improve the ability to correctly assess the degree of liver fibrosis [57]. Non-invasive biomarkers have previously been comprehensively reviewed by many studies and can be referred to by other research groups [62,63,64]. This field continues to evolve, and the search for ideal biomarkers is ongoing. Recent advances in the -omics approach have generated many clinically significant biomarkers for liver fibrosis; however, these newly identified biomarker candidates need validation in terms of performance characteristics.

#### 2.1.1. Evolving Biomarker Candidates for Liver Fibrosis

Next-generation sequencing has identified mutations in genes that are highly expressed in hepatic cells—such as *ABCB4*, *ALDOB*, *GBE1*, FAH, *ASL*, *SLC25A13*, and *SERPINA1*—which predispose the liver to fibrosis [65]. Genetic variants in non-parenchymal cells, as well as in the inflammatory cytokines, have also been shown to increase susceptibility to fibrosis [66,67]. Seven genomic loci on chromosomes 4, 5, 7, 12, and 17 that impact fibrosis phenotypes have been identified based on quantitative trait locus analysis [63]. A summary of several genetic polymorphisms implicated in the occurrence of liver fibrosis was provided by Acharya et al. [12]. Wang et al. utilized single-cell RNA-sequencing-derived data from fibrotic/cirrhotic human livers and identified 61 liver-fibrosis-associated genes that may serve as a catalog of translatable drug target candidates. Furthermore, the gene regulatory network analysis identified *CREB3L1* as a master regulator of many of these fibrosis-associated genes [64]. In the context of epigenetics, abnormal DNA methylation patterns have been found to be associated with inappropriate gene repression in liver fibrosis. Mild and severe liver fibrosis may show differential DNA methylation patterns at peroxisome proliferator-activated receptor-γ promoters in cell-free DNA [68]. Metabolomics comparison has identified serum metabolite signatures of liver fibrosis progression in chronic hepatitis C patients. Four serum metabolites were found to be significantly elevated in HCV patients with more advanced liver fibrosis severity. The choline–uric acid ratio was found to optimally differentiate between the early and late stages of liver fibrosis [66].

Numerous studies published in the last few years have established the role of urokinase plasminogen activator receptor (uPAR) and soluble urokinase-type plasminogen activator (suPAR) as biomarkers for liver fibrosis [67]. suPAR is a non-specific biomarker of inflammation. Elevated levels of it can be found in the bloodstream, and this is a strong indicator of chronic inflammation and underlying pathologies [69,70]. suPAR has also been shown to be elevated in chronic liver dysfunction, such as progressive liver fibrosis/cirrhosis [71,72,73]. Plasma levels of uPAR have been found to be closely related to the fibrosis stage in chronic hepatitis B and C [69]. Elevated suPAR concentration in cirrhotic patients is correlated significantly with the degree of cirrhosis and liver failure. Thus, serum suPAR is a potential novel biomarker for the diagnosis of cirrhosis, and indicative of an adverse prognosis (Figure 2) [74,75].

Using a cutoff level of > 9 ng/mL, suPAR predicted a poor prognosis, with a sensitivity and specificity of 70.7% and 77.8%, respectively [70]. In patients with decompensated cirrhosis, the suPAR level was significantly higher (median 12.9 ng/mL) than in patients with compensated cirrhosis (8.9 ng/mL) [76]. A strong correlation between suPAR and declining liver function (increasing AST/ALT and INR), independent of the etiology, was found [72]. suPAR has been evaluated as a prognostic marker of the severity of acute alcoholic pancreatitis. Using a cutoff value of 5 ng/mL, the sensitivity and specificity for predicting moderate or severe pancreatitis were 79% and 78%, respectively [77].

Another receptor, known as the Fas receptor or apoptosis antigen (APO-1), is a death receptor found on the surface of cells that, when bound to its ligand (the Fas ligand), induces programmed cell death (apoptosis) (FasL) [78]. Additionally, it has a soluble form (sFas) that is generated via alternative mRNA splicing. Both sFas and sFasL are non-invasive serum cell death biomarkers. sFasL levels have been found to be elevated in hepatitis, acute liver failure, and sepsis [74]. Additionally, sFasL levels are increased in NASH and steatosis [75,79]. As a result, the Fas/FasL signaling pathway is associated with a variety of diseases, including hepatitis, alcoholic liver disease (ALD), and fatty liver/steatosis associated with obesity [75].

#### 2.1.2. Role of Gut Microbiota as a Biomarker in Liver Fibrosis

Recently, many studies have disclosed the relationship between liver fibrosis and alteration of the gut microbiota and have successfully identified the gut microbiota as a biomarker for predicting liver fibrosis. Li et al. reported a lower community richness in rats with liver fibrosis in comparison to rats with normal livers. A significant difference in bacterial community diversity was found between different fibrosis stages [80]. Metagenomic phylogenetic analysis of stool samples revealed a universal gut-microbiome-derived signature that accurately identifies cirrhosis across geographically separated cohorts regardless of etiology. The study also suggested that the key microbial species within the signature might play causal roles in the pathophysiology of cirrhosis [81]. Another study reported a panel of 30 features, including 27 bacterial features, with a discriminatory ability to detect cirrhosis in patients with non-alcoholic fatty liver disease [82]. *Bacteroides* and *Ruminococcus* have been proven to be associated with NASH and the severity of fibrosis [83]. Loomba et al. demonstrated a differential gut microbiome composition of NAFLD patients with or without fibrosis [84]. Although the gut microbiome signature shows promising results to improve disease diagnosis, to realize its full potential, multicentric human studies with a large sample size are required

### 2.2. Role of Metabolic Agents

#### 2.2.1. Farnesoid X Receptor Agonist

The farnesoid X nuclear receptor (FXR), also known as the bile acid receptor, is involved in the secretion and reabsorption of bile acids. Its activation implies decrease in gluconeogenic gene expression, improved hyperglycemia, peripheral insulin resistance as well as reducing circulating triglycerides [85,86]. Hence, it is a potential target for NASH and related liver fibrosis.

Obeticholic acid (OCA) is an agonist of FXR that decreases bile acid synthesis and exerts anti-inflammatory and antifibrotic effects. In a multicenter, randomized, placebo-controlled trial [87,88] patients with NASH exhibited improvements in liver histological features after treatment.

#### 2.2.2. PPAR Agonist

PPAR is a key regulator of lipid metabolism, and is approved by the Food and Drug Administration (FDA) as a molecular target for dyslipidemia [89]. Isoforms of PPAR include PPARα, which regulates cholesterol and bile acid homeostasis, and PPARγ, which contributes to inhibiting the activation of HSCs and reduces collagen production [90]. (Lanifibranor) IVA337 is a next-generation pan-PPAR agonist that has shown preventive and curative effects on fibrosis in a CCL*4* model [91], and was tested clinically in patients without worsening the fibrosis at a dosage of 1200 mg, decreasing the SAF-A score by at least 2 points [87].

#### 2.2.3. Insulin-Based Targets

The phase 3 ARMOR study (NCT04104321) inhibits SCD1 (stearoyl-CoA desaturase-1), which promotes the synthesis of fatty acids and reduces insulin resistance. The drug has been seen to be well tolerated in animal studies as well as in phase 2 trials [88]. There was a statistically significant decrease in fat percentage in patients receiving 300 mg of Aramchol vs. placebo. The details of various targets and their modes of action are shown in Table 2, which primarily highlights some current studies affecting NASH- and NAFLD-related liver fibrosis.

#### 2.2.4. Renin–Angiotensin System Inhibitor

The renin–angiotensin system (RAS) is a crucial regulator of liver fibrosis as well as portal hypertension [98]. Activated HSCs can secrete angiotensin II, which can promote fibrosis via the angiotensin receptors [90]. Similarly, angiotensin receptor blockades may also attenuate liver fibrosis. Losartan (50 mg daily for 48 weeks) can decrease serum aminotransferase levels and promote improvements in NASH, with no adverse events [99,100]. Recently a phase 3 trial has also been posted to compare the efficacy of candesartan and ramipril in hepatitis-C-virus-related liver fibrosis [101,102]. Ras has also been shown to be associated with hypertension; therefore, candesartan—a widely used therapy—has shown promising results in clinical trials as well. It has demonstrated significant improvement in treatment outcomes with a reduction in fibrosis scores and α-SMA-positive fibrotic areas [12]. Long-term treatment with irbesartan in severe fibrosis with chronic hepatitis C showed no improvement in fibrosis score, but was well tolerated and considered to be a safe treatment [103].

#### 2.2.5. Inhibition of HMG-CoA Reductase

Statins are HMG-CoA reductase inhibitors, which reduce serum cholesterol levels by inhibiting the activity of HMG-CoA reductase [104]. The effects of statins on reducing liver inflammation, oxidative stress, and fibrosis have been reported in many animal model studies [105]. However, the safety of statins needs to be evaluated further. Moreover, a study reported that 3% of cirrhotic patients who were administered statins had severe rhabdomyolysis [106]. Currently, placebo-controlled trials examining statins, including simvastatin, are underway to check the safety of the drugs [107]. Various clinical trial details are shown in Table 3, highlighting studies associated with liver fibrosis.

### 2.3. Cellular Target-Specific Fibrosis Resolution

A targeted approach is directed towards known or established molecular targets or pathways that are critical for fibrogenesis or fibrolysis and, more importantly, do not overlap significantly with unrelated pathways, so as to avoid potential side effects [114]. These agents may be enzyme inhibitors or small-molecule inhibitors.

This review article discusses a variety of lesser-known transcription factors (Table 4). Numerous antifibrotic strategies have been developed to inhibit the pro-fibrogenic TGF signaling pathway, including the use of soluble TGF receptor type II [115,116], TGF-blocking antibodies [117], and TGF antisense oligonucleotides, or molecules [118] that disrupt downstream signal transduction. Lerdelimumab and metelimumab—two monoclonal antibodies against TGFβ—are currently in phase 3 and phase 1/2 clinical trials for reducing scarring after glaucoma surgery and systemic sclerosis, respectively [119,120,121]. However, systemic targeting of the TGF pathway is limited because, in addition to stimulating wound healing and fibrosis, it acts as a central inhibitor of inflammation, and is required for epithelial differentiation and apoptosis. Another promising strategy is to inhibit tissue inhibitor of metalloproteinase-1 (TIMP-1) [122]—a major mediator of liver fibrosis—using a recombinant mutant protein derived from its ligand MMP-9 [123].

An additional intriguing novel target is integrin v6—a cell surface receptor expressed on activated epithelia during wound healing, tumorigenesis, and embryogenesis [124,125]. It is expressed exclusively on activated cholangiocytes in the liver, which act as potent promoters of liver fibrogenesis [126]. Inhibition of v6 has been shown to reduce collagen deposition, improve liver function, and slow the progression of fibrosis [127]. Hepatocyte nuclear factor 4 (HNF4) is another transcriptional factor involved in the differentiation and function of hepatocytes. Forced expression of HNF4 has been shown to ameliorate hepatic fibrosis, improve liver function, and inhibit EMT in a fibrosis model [128]. Additionally, siRNA-mediated inhibition of HNF exacerbated hepatic fibrosis and decreased E-cadherin, vimentin, and fibroblast-specific protein-1 expression [129].

**Table 4 cells-11-01500-t004:** Molecular targets for antifibrotic therapies and their associated clinical trials.

Target	Cells	Drug	Clinical Trial Stage	Clinical Trial Details	References
TGFβ	HSCs, cholangiocytes, inflammatory cells, endothelia	Soluble type II, anti-TGFβ antibody;	Phase 2	FG-3019 for HBV infection, anti-CTGF monoclonal antibody;	[119,120,121]
		Lerdelimumab;	Phase 3	Monoclonal antibody, Cambridge antibody technology;	
		Metelimumab	Phase 1/2	Monoclonal antibody, Cambridge antibody technology	
TIMP1	HSCs, human endothelial cells, lymphoma, and breast carcinoma	MMP antagonist		Serum levels of TIMP-I in 268 patients with liver diseases	[122]
Integrin αvβ6	Activated epithelia	Small-molecule antagonist, blocking AB	Phase 2	Monoclonal antibody (STX-100) for idiopathic pulmonary fibrosis	[125]
TLR-4	Macrophages, HSC	Small-molecule antagonists and downstream targets	Preclinical trial	TLR4-deficient mice protected against hepatic injury; vaccine development for hepatitis B (GlaxoSmithKline/Dynavax)	[130,131]
HNF4α	Hepatocytes, pancreatic beta cells	HNF4α agonists	Preclinical trial	(CureVac) restoration of HNF4α via mRNA delivery using paraoxonase 1 as a therapeutic target	[132]
LPA	HSC	LPA receptor and small-molecule antagonist	Phase 2	Idiopathic pulmonary fibrosis treatment using the LPA1 pathway	[133]

Lysophosphatidic acid (LPA) is a lipid mediator that has been implicated in a variety of functions, including apoptosis, migration, proliferation, and cancer cell invasion [134]. The expression of LPA and LPA1R (lysophosphatidic acid receptor) is elevated in a variety of inflammatory states [135]. While LPA has a variety of physiological effects on the receptors of parenchymal cells, LPA1R antagonists have been shown to have an antifibrotic effect on liver fibrosis and lung fibrosis [133,136,137]. Table 4 summarizes the molecular targets, as well as the clinical trials that have been conducted and their current statuses.

### 2.4. Role of Growth Factors in Liver Fibrosis

HGF was identified as a mitogen for hepatocytes; it is produced by stromal cells, and stimulates epithelial cell proliferation, motility, morphogenesis, and angiogenesis in a variety of organs [138]. TGF-β plays a critical role in tissue fibrosis during chronic organ injury by converting HGF-producing fibroblasts to ECM-producing myofibroblasts [139,140]. HGF inhibits TGF-β production in myofibroblast cultures [141,142], and also blocks the TGF-mediated signaling pathway by inhibiting nuclear Smad2/3 activation [143], which can result in antifibrotic effects *in vivo*. Additionally, HGF inhibits the function of other fibrotic cytokines such as PDGF and CTGF/CCN2 and acts as a pro-fibrogenic factor, the details of which are mentioned below. MMPs were also induced in myofibroblasts by HGF [144,145]. MMPs must be induced by HGF not only for ECM degradation, but also for myofibroblast elimination, both of which contribute to fibrosis resolution. For example, HGF-induced MMP 9 degrades fibronectin—a critical cellular anchor [144]; this results in the apoptosis of myofibroblasts, which is required for the resolution of fibrosis. The resolution of fibrosis creates an opening for epithelial and endothelial repair, which may result in organ recovery. HGF is also required for tissue protection during inflammatory diseases, either directly on macrophages, dendritic cells, or lymphocytes [138] (immunogenic-cell-based mechanism), or indirectly on epithelial cells (epithelial-cell-based mechanism). Both mechanisms have the potential to be beneficial in inflammatory states. Given that HGF is required for organ protection and tissue regeneration, it is reasonable to assume that HGF-based therapy, variants, or fragments—in combination with activation of HGF/c-Met signaling, which decreases TGF- mediators—may show promise in treating a variety of inflammatory or fibrotic diseases [146]. Figure 3 shows a balance between MMPs and TIMPs for potential fibrosis resolution.

CTGF/CCN2 is a pro-fibrogenic molecule and a multifunctional matricellular protein produced by a variety of cell types. Of the many functions of CTGF/CCN2, it also has the ability to promote fibrosis, and can be seen to be overexpressed in many fibrotic lesions, including in the liver [147]. CTGF/CCN2 is activated by TGF-β, and mediates ECM-inducing properties previously attributed to TGF-β. In the fibrotic liver, CTGF/CCN2 mRNA and protein are produced by fibroblasts, myofibroblasts, HSCs, endothelial cells, and bile duct epithelial cells [148]. CTGF/CCN2 expression in cultured HSCs is also enhanced following their activation by TGF-β, while exogenous CTGF/CCN2 promotes HSC adhesion, proliferation, and collagen production. CCN2’s action has also been confirmed *in vivo*, with a transgenic FVB mouse [148]. Production of human CCN2 mRNA and its elevated levels were found in transgenic livers. It is believed that during the initiation of downstream fibrogenic events in the liver, the production of CTGF/CCN2 is regulated primarily by TGF-β, and CTGF/CCN2 plays an important role in HSC activation and the progression of fibrosis [147].

Another growth factor that works via the PDGF-α receptor and is considered to be a potent mitogen for human fibroblasts, as well as vascular smooth muscle cells in vitro, is the PDGF [149]. Studies suggest that this also plays an important role in the regulation of fibrosis. Initial experiments included a murine model derived from the transgenic overexpression of PDGF in the heart-induced fibroblast proliferation, which ultimately resulted in cardiac fibrosis, hypertrophy, and eventually cardiomyopathy [150,151]. Transgenic mice with liver-specific PDGF overexpression experienced liver fibrosis and hepatocellular carcinoma [152]. Blocking PDGF signaling has also been known to inhibit HSC proliferation and ameliorate liver fibrogenesis [153]. Clinical studies have also shown that excessive activation of PDGF and its downstream molecules is associated with necroinflammation and fibrosis in patients with hepatic damage [154,155]. Hence, PDGF and its signaling pathway play an important role in the development and prognosis of hepatic fibrosis.

### 2.5. Reduction in Inflammation and Immune Response

Inflammation is a critical and complex component of liver fibrosis; following liver injury, an accumulation of recruited inflammatory cells occurs at the site of the injury [20]. Platelets, neutrophils, macrophages, mast cells, and NK cells from the innate immune response, as well as T and B cells from the adaptive immune response, also participate in the fibrogenesis process [156]. The inflammatory response of immune cells during the process of fibrosis is mediated by a cocktail of pro- and anti-inflammatory compounds, including cytokines, chemokines, and growth factors [157,158]. HSCs also actively participate in the inflammatory process through their interactions with various immune cells. Additionally, HSCs are converted from a dormant state to an activated state via the myofibroblast-like phenotype, which is involved in proliferation and extracellular matrix deposition [159].

Tumor necrosis factor-alpha (TNF-α)-stimulated gene 6 (TSG-6) was identified as a cDNA derived from TNF-treated human fibroblasts, which maps to chromosome 2q23.3 [160,161,162,163]. This 35 kDa glycoprotein is not found in healthy adults, but is produced in response to inflammatory mediators, and is detected in a variety of inflammatory diseases, including those that cause liver fibrosis [164]. However, it has been demonstrated that increased TSG-6 expression during an inflammatory process contributes negatively to the inflammatory response [165]. TSG-6 was recently identified as a critical immune modulator secreted by human mesenchymal stem cells (hMSCs), and is responsible for hMSC-based therapeutic effects such as improved wound healing and cardiac function [166] but, more importantly, its effect on liver regeneration using conditioned media and organoids derived from TSG-6-treated HSCs in an acute liver injury model has been demonstrated [167]. In recent years, studies have documented that signal transducer and activator of transcription 3 (STAT 3) is closely related to the development and occurrence of liver fibrosis. TSG-6 was found to be effective at inhibiting hepatic oxidative stress and inducing hepatic M2 macrophage polarization by suppressing STAT3 activation [168]. Additionally, this study suggests that TSG-6 may play a critical role in liver regeneration, and may act as a protective factor against liver damage caused by inflammation and fibrosis [168], as illustrated in Figure 4. This could aid in the development of more novel therapeutic strategies, either alone or in combination.

### 2.6. Immune-Mediated Role of NK Cells

Cell senescence is a state of terminal growth arrest that is triggered by stress signals and cellular damage, and ultimately results in cell apoptosis if not corrected [169]. Senescent cells have an enlarged morphology and distinct metabolic and gene expression patterns in comparison to proliferating cells. Additionally, they exhibit a senescence-associated secretory phenotype, which is characterized by the production of pro-inflammatory factors, angiogenesis factors, and MMPs, which alter tissue function by promoting angiogenesis, attracting immune cells, and remodeling the ECM [170,171,172]. Due to mounting evidence that senescent cells have a detrimental effect on age-related declines and inflammatory diseases such as fibrosis, a major goal in this field is to develop an intervention capable of selectively identifying and eliminating senescent cells. Because these cells express a wide variety of proteins—particularly those on their cell surface—the concept of identifying the presence of a unique cell surface protein on senescent cells is critical in this strategy. Dipeptidyl peptidase 4 (DPP4 or CD26) was identified as a surface protein expressed more abundantly in senescent cells using mass spectroscopy [171]. RNA isolation from proliferating and senescent cells, followed by RT-qPCR analysis, also revealed that senescent cells expressed significantly more DPP4 mRNA, implying that DPP4 may promote cellular senescence [171].

NK cells are innate immune system lymphocytes that can rapidly eliminate stressed or apoptotic cells [173]. NK cells kill senescent cells via exocytosis of granules containing perforin and granzyme, and produce IFN-γ in response to senescent cell interaction [174]. Another preferential method of eradicating senescent cells is through antibody-dependent cytotoxicity (ADCC) [175]. The ADCC assay employs antibodies to recognize specific antigens on the cell surface, and directs natural killer cells to selectively destroy antibody-labeled cells [176]. The presence of DPP4 on the surface of these cells makes them suitable targets for NK cells that recognize anti-DPP4 antibodies, and the dual action of DPP-induced cellular senescence and NK cells can be seen in action in Figure 5 [171]. Additionally, it was observed that using an anti-PD1 antibody (CT-011) enhanced NK cell function [177]. It has been demonstrated that inhibiting the PD-1 pathway enhances NK cells’ IFN-γ release.

### 2.7. Role of Cytokines in Liver Fibrosis

Suppressive cytokines are critical for orchestrating the shape of numerous immune cells, along with fibrotic tissue reduction. Interleukins (IL) are a family of immunomodulatory cytokines or small signaling proteins that play a critical role in immune response regulation [178]. They are produced by a variety of cell types during inflammatory responses, and the balance of these cytokines dictates the outcome of the immune response. As a result, they are a critical therapeutic tool in the treatment of patients with liver diseases. In chronic liver diseases, interleukins may have both pro- and anti-inflammatory functions; some even express both, and are dependent on the inflammatory stimulus [178]. For example, IL-17 can promote hepatic fibrogenesis by activating hepatic stellate cells, whereas IL-22 protects against the development of fibrosis or steatohepatitis [179]. Similarly, IL-13 and IL-33 are related to Th2 and innate lymphoid cells, respectively, and contribute to the fibrotic response to liver injury, whereas IL-10 is a model anti-inflammatory interleukin with tissue-protective properties during chronic liver injury and fibrogenesis [178]. Table 5 summarizes the functions of various interleukins.

Several cytokines can affect the number as well as the function of myofibroblasts. Both inhibitory and stimulatory effects on myofibroblasts have been described [191]. TGF-β, PDGF and IL-6 are key cytokines for the formation and activity of myofibroblasts. Additionally, IL-4, IL-13, and IL-22 show pro-fibrotic activity by enhancing the production of collagen type I in normal fibroblasts [192,193,194]. TNF-α induces proliferation and collagen synthesis of atrial and intestinal myofibroblasts [195]. Apart from these stimulatory cytokines, several signaling molecules inhibit myofibroblast formation and activity. For instance, interferon-γ (IFN-γ) inhibits collagen synthesis, sensitizes skin fibroblasts to Fas-mediated apoptosis, and inhibits the effects of IL-4 [196,197]. A time- and dose-dependent induction of αSMA expression in human lung fibroblasts has been demonstrated, suggesting that myofibroblasts can be partially activated and enhance the production of collagen type I in normal fibroblasts [194,198]. Additionally, cytokines have the potential to enhance immune-cell-mediated immunity, particularly in NK cells. Due to its ability to promote T- and NK-cell proliferation, homeostasis, and cytotoxicity [199], IL-2 was the first cytokine to be used in clinics to enhance immune responses, including a novel IL-2 variant called super-2 with a higher binding affinity for IL-2R [200]. This modified IL-2 induced an increase in cytotoxic T-cell expansion and a decrease in Treg-cell activation. Another cytokine, IL-15, stimulates CD8+ T cells and non-differentiated NK cells, suggesting that it could be a significant immunotherapeutic agent [201]. IL-12 is another cytokine that can be administered to enhance NK cells’ cytolytic activity. It facilitates the release of IFN-γ, migration, and NK-mediated ADCC (as mentioned in Section 2.6) [202].

Chemokines are a family of small heparin-binding molecules that assist leukocytes in infiltrating the liver following acute or chronic injury [203]. Chemokines are produced by resident cells of the liver, including hepatocytes, HSCs, leukocytes, and platelets. Recently, numerous significant effects of specific chemokines and their receptors have been discovered. The C-X-C chemokine receptor type 3 (CXCR3) is a critical chemokine receptor that binds to the CXC chemokine ligands CXCL9, CXCL10, and CXCL11 [204]. CXCL9 is secreted directly by hepatocytes, and can bind to HSCs via the CXCR3 receptor. Unlike other chemokines, CXCL9 does not stimulate HSCs, but inhibits collagen secretion and mRNA expression. Given the critical role of stellate cells in liver fibrogenesis, it is reasonable to assume that CXCL9 has antifibrotic properties [203,205].

### 2.8. Role of miRNA Family

The mechanism of hepatic fibrosis is a highly complex process involving numerous cellular and molecular events. As a result of exogenous factors, dormant HSCs become activated and transform into myofibroblasts, resulting in the formation of ECM and, ultimately, liver fibrosis. The progression of hepatic fibrosis is associated with a variety of integrated signaling pathways, including the MAPK, Wnt, PI3K/AKT, and Hedgehog/Gli pathways [206]. The use of miRNAs to induce RNA interference and thus regulate gene transcription levels, is generating considerable interest in this field. miR-29 has been an extensively studied and referenced microRNA in the development of liver fibrosis [207]. It is a miRNA that inhibits fibrogenesis and is also required for HSC activation. The important fibrogenic cytokine TGF-β is a potent activator of HSCs [208]. It inhibits miR-29 expression and promotes a fibrogenic environment by activating HSCs and increasing ECM deposition [209,210]. Additionally, by modulating the PI3K/AKT pathway, the miR-29 family induces cell apoptosis [206]. By inhibiting alpha-1 type I collagen, miR-29 overexpression results in decreased collagen deposition (Col1A1) [209,211]. Additionally, this family is involved in post-translational ECM and fibril formation processing. The miR-29a subgroup has been shown to inhibit the activation of qHSCs by targeting HDAC4, whereas miR-29b induces cell apoptosis by targeting AKT3 and PI3KR1 [212]. Zhang et al. established an inverse correlation between miR-29b expression and heat shock protein (HSP47) expression, which are critical regulators of ECM maturation, and demonstrated that miR-29b overexpression results in abnormal collagen formation [213].

Similarly, it has been demonstrated that the miR-15 family promotes cell proliferation and induces apoptosis [214,215]. As illustrated in Figure 6, liver cells take and release exosomal miRNAs, and the extracellular vesicles (EVs) that are derived from adipose tissue, HSCs, and neutrophils [216]—containing miR-155, 214, and 223, respectively—are taken up by the liver cells, which can lead to suppression of PPARγ, leading to insulin resistance [217]. The subgroup miR-16 targets HGF and Smad7 to promote the TGF-/Smad signaling pathway, and inhibits cell proliferation by targeting CD1, whereas miR-15b and miR-16b promote apoptosis by targeting Bcl-2 [206]. Yan et al. identified a novel miRNA (miR-34a) that promoted HSC activation by targeting acyl-CoA synthetase long-chain family member 1 (ACSL1). Inhibiting miR-34a increased ACSL1 levels and decreased α-SMA levels [218]. Yuan et al. investigated the therapeutic potential of miR-155 in mice following liver injury caused by N-acetyl-p-aminophenol (APAP). It was found to be upregulated in liver tissue and blood samples following APAP injury. miR155-/- mice had elevated AST and ALT levels, as well as inflammatory mediators such as TNF-α and IL-6. Its deficiency may result in increased p65 and IKK expression, activating NF-κB [219]. Additionally, connective tissue growth factor (CTGF/CCN2) has been shown to be associated with fibrogenesis in HSCs. CCN2 upregulation is associated with miR-214 downregulation in fibrotic or steatotic livers or culture-activated primary murine HSCs [220]. miR223 was also found to protect against liver fibrosis via its regulation of multiple hepatic cell targets. It inhibits the transcriptional coactivator with PDZ-binding motif (TAZ) expression in hepatocytes, resulting in a decrease in hepatocyte-derived Indian hedgehog (IHH) secretion, which acts as a ligand for hedgehog signaling in HSCs [221].

### 2.9. Role of Mesenchymal Stromal Cells in Liver Tissues

MSCs are one of the main candidates for cellular therapies. Most of the time, these cells come from adult bone marrow (BM-MSCs), but umbilical cord blood (UCB) has also been shown to be a good source of hematopoietic stem cells, with evidence that MSCs do exist in CB [222]. Since they can change into mesodermal cells, they have a wide range of immunomodulatory properties. Many other immune cells can also be taken from the cord blood mononuclear cells (CBMCs), including the NK cell population [223], but the number of CB NK cells is very low, and they are known to be incompletely formed and need to be activated to keep working properly [224]. There are many ways to increase the number of these cells, which could help the immune system return to normal after a direct cell transplant, as well as in other cell-therapy-based research areas.

MSCs are also considered to be immune-privileged because they lack class II major histocompatibility complex (MHC) molecules, and have a very low number of class I MHC molecules, which enables their allogeneic use [225]. Numerous studies in clinical trials indicate that transferred MSCs have an antifibrotic effect (Table 6), but the timing of the therapy is critical, as the effect on fibrosis is mediated by a reduction in inflammation rather than directly promoting degradation. Numerous studies have been published demonstrating a decrease in TGFβ-1 and α-SMA gene expression in liver cells following MSC treatment [226,227,228,229]. Administration of BM-derived MSCs alleviated fibrosis and improved the hypoxic liver microenvironment in a CCL4-induced animal model. These cells exhibit a direct relationship with the TGF2/SMAD signaling pathway in liver cells [230,231]. Several studies also support the idea of MSCs being engrafted/differentiated directly into damaged tissue. The engrafted cells were detected long after transplantation in a liver intoxication model caused by a lethal dose of APAP [232].

MSCs can also exert pro-regenerative and antifibrotic effects in liver tissues by inducing the proliferation of resident mature hepatocytes or progenitor cells via the secretion of paracrine factors. They can inhibit the activation of HSCs in vitro via the production of IL-10, VEGF-A, and HGF, as demonstrated by gene expression analysis [243]. Additionally, the MSC secretome has been shown to be less invasive and effective in liver regeneration [244]. Numerous studies demonstrate that the secretome obtained from UC-MSCs differentiated or committed to hepatocyte-like cells enhances hepatic function both in vitro and in vivo. UC-MSC enriched with milk factor globule EGF8 (MFGE-8)—an antifibrotic protein—suppressed the expression of fibrosis by downregulating the α-SMA and TGF-β pathways [245]. TGF-activated HSCs using conditioned media from amniotic MSCs (AMSCs) [246] and BM-MSCs [247] also exhibited antifibrotic activity

EVs have been studied recently for their potential role in disease. MSC-derived EVs have been shown to promote tissue regeneration in a variety of tissues, including the heart, lungs, brain, kidneys, and liver. In a variety of preclinical models [248,249], MSC-EVs have been shown to possess therapeutic properties. In terms of fibrosis, they act on hepatocytes, activated HSCs, and immune cells by modulating their signaling pathways. Li et al. [250] provided one of the first pieces of evidence demonstrating that EVs from UC-MSCs alleviate hepatic inflammation and collagen deposition in a CCL4 liver fibrosis model. TNF-β, IL-1, and IFN-γ mRNA expression in liver tissue was decreased by UC-MSC-EVs [251]. Embryonic stem cells (ESCs) have also been identified as a potential alternative source of MSCs, with ESC-MSC-EVs increasing hepatocyte viability and decreasing apoptosis and pro-fibrotic cytokine expression in a thioacetamide (TAA) animal model [252]. Thus, MSC treatment has been shown to reduce fibrosis by downregulating fibrosis-related gene expression levels, including α-SMA, TGF-β, and collagens, with promising results obtained in preclinical studies using MSC-EVs.

### 2.10. Current Challenges in Clinical Trials

The current progress and research being conducted to better understand the mechanisms of hepatic fibrosis and its therapeutic targets emphasizes the critical nature of developing clinical trial designs that can evaluate the efficacy of antifibrotic drugs. It is also critical to understand the dynamics of fibrosis regression and the current inaccuracies of standard fibrosis staging systems (e.g., Ishak, Brunt, Metavir) [253], such as collagen proportionate area quantification [254]. Sustained suppression of hepatitis B and C results in significant improvements in inflammation and necrosis, as well as regression of fibrosis. Challenges are also directly related to drug efficacy. Selecting the appropriate endpoint and duration of therapy are also essential for assessing the efficacy of the drug in clinical trials. The reversal of NAFLD/NASH (without worsening fibrosis) or the improvement of fibrosis (with no further deterioration of NASH) are the endpoints for pre-cirrhosis patients. Moreover, adequate stratification is essential to ensure reliability in clinical trials. In trials for NAFLD, similar drugs are often given to patients with different underlying comorbidities, which can lead to a variety of treatment responses that must be considered and managed with specific strategies.

At the moment, many new antifibrotic agents being tested in clinical trials focus on NASH as an etiology [255]. This increased awareness and focus on NASH has resulted in remarkable advancements in specific therapies, as well as a growing understanding of obesity- and fatty-liver-related diseases, which affect 10 times as many people as HCV in the United States and Europe [256]. Another factor that could be considered is the stage of the disease—specifically, drugs that target inflammation and cell injury. They may be effective at the disease’s early and intermediate stages. Innovative trial designs should be considered, as they may help to address current pitfalls associated with liver fibrosis trials, including NASH and NAFLD. Ultimately, the approval of antifibrotic drugs will be based on endpoints that are either directly related to or reasonably predict specific clinical outcomes.

## 3. Conclusions

We continue to gain a better understanding of the pathophysiology of liver fibrosis. It can be triggered by genetic and metabolic disorders, chronic viral hepatitis, infections, drugs, cholestasis, and other environmental factors. Medical complications can occur because of the accumulation of ECM, disruption of lobular structure, and deterioration of hepatocellular function during this process. Numerous cellular and extracellular agents have been identified that can be activated or transformed into ECM-synthesizing phenotypes. Numerous experimental studies have demonstrated that a variety of agents may have antifibrotic potential. Despite this, only a small number of candidates have progressed successfully to the clinical trial stage. Numerous potential therapies have emerged in the current scenario that demonstrate promise—either individually, or as part of a combination study that targets multiple cells or pathways in order to provide a more holistic solution to the current fibrosis issues. However, while numerous therapies have demonstrated promise, additional research is necessary to determine whether these therapies can be translated into clinical practice. At the moment, removing fibrosis-causing agents and factors affecting stellate-cell activation remains a critical strategy for reducing and preventing the disease.

## Figures and Tables

**Figure 1 cells-11-01500-f001:**
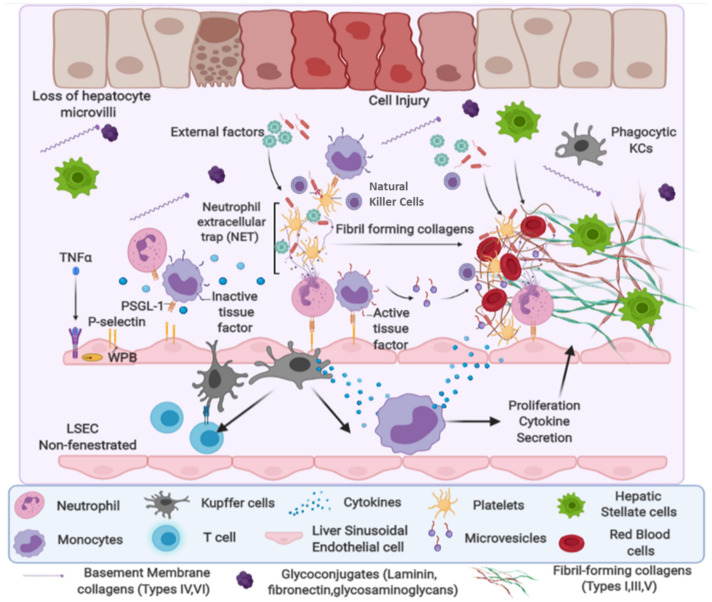
Cellular alterations during hepatic fibrosis: Multiple factors can cause liver injury. These factors induce liver inflammation through various pathways and cellular systems. Normal liver parenchyma contains a hepatocyte layer with microvilli and layers of fenestrated liver sinusoidal endothelium cells (LSECs), and a higher number of quiescent hepatic stellate cells (qHSCs), Kupffer cells (KCs), and natural killer cells (NKs). It also contains a normal amount of basement-forming collagens (Types IV and VI). Upon injury, the HSCs become activated and secrete a large amount of ECM, which leads to the loss of both endothelial fenestrations and hepatocyte microvilli, resulting in impairment of bidirectional metabolic exchange of portal venous flow. TNF-α can mediate a dual and opposing effect by acting on TNF receptors expressed on the endothelial cells. The LSECs also promote vascular leakage of plasma proteins and initiate the exocytosis of Weibel–Palade bodies (WPBs, denoted as a yellow oval), bringing P-selectin to the cell surface, which initiates diapedesis. Replacement of fibrillary collagen occurs, consisting of collagen I, III, and fibronectin. Furthermore, there is infiltration of immune cells, such as neutrophils and monocytes, and the injured area recruits the NK-T cells and alters liver morphology.

**Figure 2 cells-11-01500-f002:**
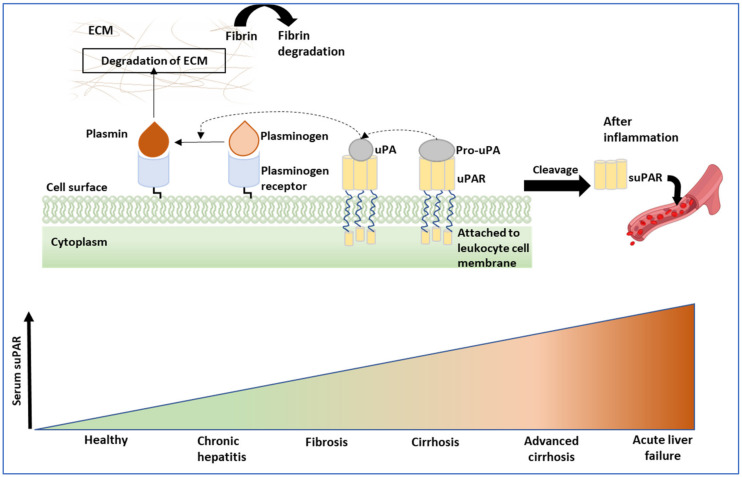
suPAR as a serum biomarker: Upon inflammation and for controlling fibrin degradation, the uPAR is cleaved from the cell surface as Pro-uPA and activates in the uPA form. Plasminogen is converted to plasmin via either the plasminogen activator receptor or the urokinase receptor and helps in the degradation of the ECM by breaking fibrin strands during liver fibrosis. In patients with liver diseases, circulating suPAR levels increase with the increase in disease severity, and are indicative of an adverse prognosis.

**Figure 3 cells-11-01500-f003:**
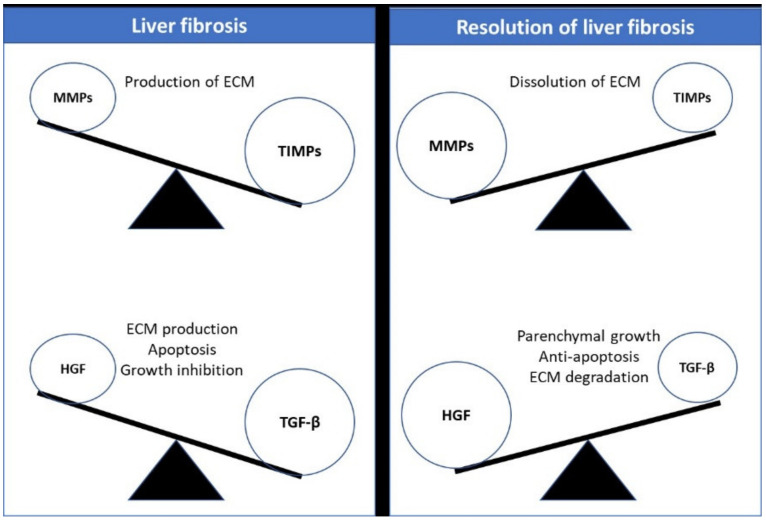
The importance of balance between MMPs and TIMPs, and between HGF and TGF-β, as hepatoprotective and counteracting agents in liver fibrosis.

**Figure 4 cells-11-01500-f004:**
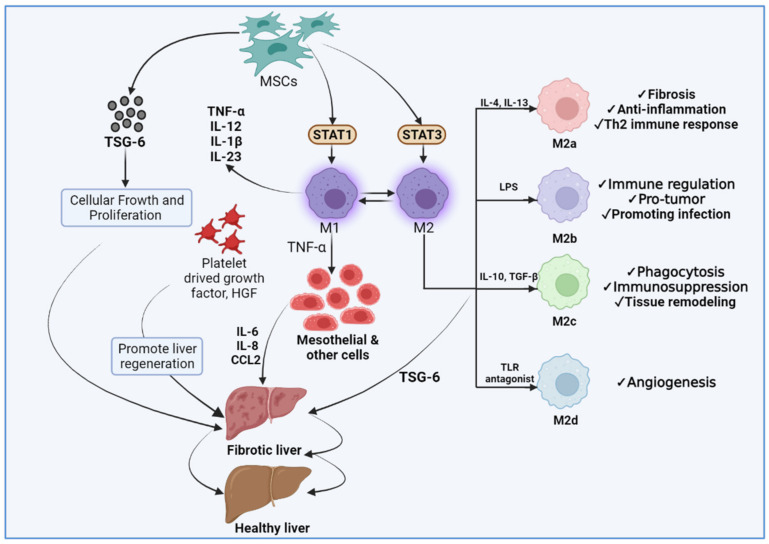
The role of TSG-6 in cellular growth and proliferation in fibrotic liver cells: TSG-6 has the potential to improve liver injury; it can induce proliferation, stemness, and increase the immunomodulatory mechanism of MSCs. TSG-6 reduces inflammation and changes in tissue repair via mechanisms such as reducing neutrophil infiltration and activation, and inhibiting inflammatory M1–M2 polarization of monocytes. M2 macrophages produce complex cytokines, and have various functions; they can be further divided into M2a, M2b, M2c, and M2d subtypes. M2a cells can prevent fibrosis by inducing regulatory T cells.

**Figure 5 cells-11-01500-f005:**
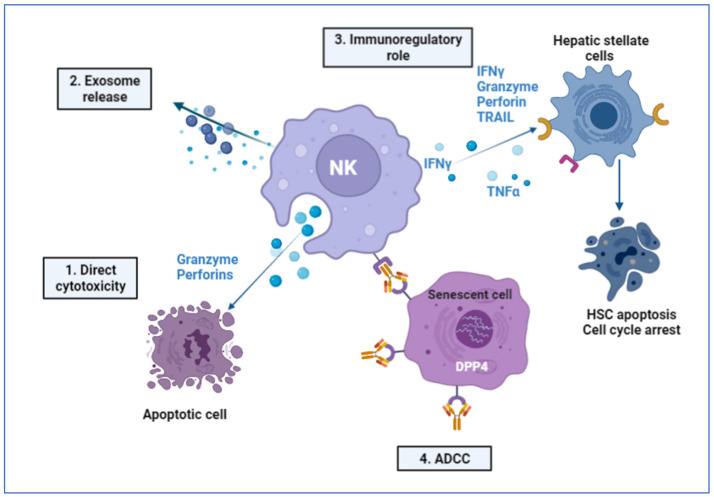
NK cell responses and targeted therapy: (**1**) NK cells are the first responders in the immune system, and can directly recognize and begin the cell death mechanism. (**2**) NK cells release exosomes with cytotoxic capabilities, and can contain miRNAs, cytokines, and NK cell surface receptors. (**3**) Activated NK cells selectively kill early or activated HSCs, but not quiescent HSCs. IFN-γ-producing NK cells directly induce HSC death, but also further enhance NK cell cytotoxicity against HSCs. Quiescent cells do not express elevated NK-activating ligand, and are hence resistant. (**4**) In proliferating cells, DPP4 is expressed at low levels, but in senescent cells, DPP4 mRNA levels increase, leading to the production of DPP4, which localizes on the cell surface and is exposed to the extracellular space. The localization of DPP4 enables selective elimination by immune cells such as NKs via ADCC.

**Figure 6 cells-11-01500-f006:**
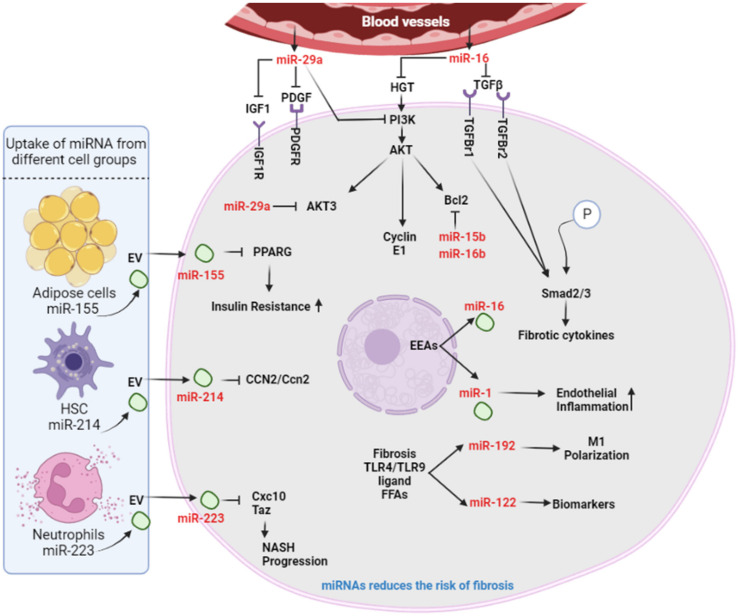
The role of various miRNA families in liver fibrosis: Liver cells take and release exosomal microRNAs (miRNAs). Their role in liver fibrosis: Extracellular vesicles (EVs) derived from adipose tissue, HSCs, and neutrophils containing miR-155, -214, and -223, respectively, are taken up by liver cells, leading to increased insulin resistance by suppressing PPARγ. miR-214 in HSCs is shuttled by EVs to hepatocytes, resulting in inhibition of CCN2/Ccn2. Under a high-fat diet and alcohol consumption, miR-223 is elevated in hepatocytes, and attenuates NASH progression by targeting Cxcl10 and Taz. The miR-29 and miR-15 families regulate hepatic fibrosis in the following ways: (1) miR-29a targets AKT3 and PI3K, which helps to induce cell apoptosis through the caspase-9 cascade pathway; (2) PDGF and IGF receptors suppress the overall effect of the PI3K/AKT signaling pathway; (3) miR-16 targets HGF and SMAD7, and blocks the TGF-β/Smad signaling pathway; (4) miR-192 downregulates cyclin M1 and inhibits cell proliferation; (5) miR-1 promotes endothelial inflammation.

**Table 1 cells-11-01500-t001:** Hepatic stellate cell phenotypes.

	Cell Type	Functions	References
1	Quiescent	Storage of vitamin A, which is found in numerous intracellular droplets. Multiple thorn-like cytoplasmic extensions can protrude into the sinusoidal space or make direct contact with hepatocytes. These extensions can also be used as sinusoidal sentinels, which can detect biochemical or mechanical alterations in hepatocytes.	[27]
2	Activated	They lose lipid-rich granules and transdifferentiate into α-SMA-positive myofibroblasts. They produce increased amounts of ECM and pro-inflammatory as well as pro-fibrogenic cytokines, and cause liver fibrosis.	[28]
3	Inactivated	Reverted or inactivated HSCs present a restored expression of their pro-fibrogenic protein profile (including changes in collagen-1, α-SMA, TGF-beta receptor type-1 (TGFRI), and TIMP1 expression). They do not express quiescent makers (such as perilipin 2 and adiponectin receptor 1).	[27]
4	Senescent	Senescent hepatic stellate cells display decreased collagen production and proliferation. Induction of senescence could be a protective mechanism against the progression of liver fibrosis. The concept of therapy-induced senescence has been proposed to treat liver fibrosis.	[29]

**Table 2 cells-11-01500-t002:** Clinical trials for metabolic targets affecting NASH/NAFLD-based liver fibrosis.

CT No.	Treatment/ Drug Name	Mechanism of Action	Target Diseases	Clinical Trial Design	Clinical Phase	Efficacy	Number of Patients	References
NCT00760513	OMACOR	Reduce the synthesis of triglycerides (TGs)	NAFLD	Treatment with long-chain n-3 fatty acid for 18 months affects biomarkers of NAFLD.	Phase 4	20% decrease in liver fat	103 participants	[92]
NCT02030977	Resveratrol	Antioxidant	NAFLD, Liver Fibrosis	Effect of liver enzymes, inflammatory factors, and fibrosis in patients with NAFLD. Patients were steatosis grade 1.	Phase 3	Study completed/ no results reported	50 patients	[93,94]
NCT02548351	Obeticholic acid REGENE-RATE	FXR agonist	NASH with fibrosis	Obeticholic acid treatment compared to placebo on histological improvement and liver-related clinical outcomes.	Phase 3	Active	2480 participants	[95,96]
NCT03008070	IVA337	PPAR agonist	NASH with fibrosis	A next-generation pan-PPAR agonist for the pathophysiology of NASH.	Phase 2	Not worsening fibrosis at higher dose of 1200 mg	247 participants	[91]
NCT02684591	Aramchol	SCD1 inhibitor	NASH	To test the efficacy of 400 mg and 600 mg of Aramchol.	Phase 2	No significant adverse effects, did not reduce hepatic fat	247 participants	[88]
NCT03357380	Semaglutide	reduces HbA1c,	NAFLD	Comparing changes in early-stage scar tissue and fat deposition in the liver. Participants self-inject medicine once daily for 72 weeks.	Phase 1	No significant adverse Effect, did not reduce hepatic fat	67 participants	[97]

**Table 3 cells-11-01500-t003:** Clinical trials for metabolic targets affecting liver fibrosis.

NCT No.	Treatment/Drug Name	Mechanismof Action	Target Diseases	Clinical Trial Design	Clinical Phase	Efficacy	Number of Patients	References
NCT00049842	Peginterferon alph-2b (SCH 54031)	Type 1 interferon activator	Liver fibrosis, chronic hepatitis C	Evaluate safety and efficacy of PEG-Intron vs. no treatment	Phase 3	Lower fibrosis progression	540 participants	[108,109]
NCT01938781	Entecavir, Peg-IFN	Inhibits HBV DNA polymerase	Liver fibrosis	For patients with F2/F3, one arm is entecavir for 2 years, and the other is entecavir for 0.5 years and entecavir plus peg-IFN for 1 year	Phase 4	Study completed/ no results reported	400 participants	[110,111,112]
NCT00298714	Losartan	Angiotensin II type 1 receptor antagonist	Liver fibrosis, chronic hepatitis C	Administration of angiotensin II type 1 (AT1) receptor antagonists in HSCs (fibrosis F2-F3)	Phase 4	Study completed/ no results reported	20 participants	[99,100]
NCT00990639	Candesartan andramipril	Angiotensin II type 1 receptor antagonist	Liver fibrosis extent with chronichepatitis C	Evaluating drug action and changes in FibroScan recording	Phase 3	Pending	45 participants	[101,102]
NCT00265642	Irbesartan	Angiotensin II type 1 receptor antagonist	Liver fibrosis, chronic hepatitis C	AT1 receptor antagonists of angiotensin II have inhibitory effects on TGF-beta 1 production, and can limit the progression of liver fibrosis	Phase 3	No results reported	200 participants	[103]
NCT04971577	Simvastatin	HMG-CoA reductase inhibitors (statins)	Liver fibrosis	Simvastatin for reducing liver fibrosis in patients with advanced fibrosis due to alcohol	Phase 2/3	Active	90 participants	[107,113]

**Table 5 cells-11-01500-t005:** Review of different interleukin functions in liver fibrosis.

Interleukin Type	Produced by	Response Cell	Function	References
**IL-2**	CD4+ T cells, CD8+ T cells, dendritic cells, and thymic cells	T cells and NK cells	Enhances cytotoxicity in NK cells; activates STAT1, STAT3, and STAT5.	[180]
**IL-6**	Lymphocytes, monocytes, fibroblasts, vascular smooth muscle cells, and endothelial cells	Non-parenchymal cells	Deletion of IL-6 increases hepatocyte injury and apoptosis.	[181]
**IL-10**	Hepatic stellate cells, liver sinusoidal endothelial cells, Kupffer cells, lymphocytes, and Th cells	HSCs	IL-10 inhibits HSC activation.	[182,183]
**IL-12**	Macrophages, dendritic cells, and B lymphocytes	Th1	IL-12 shifts immune response to Th1; Secretion of IFN-γ and augmentation of the cytolytic activity.	[35,184,185]
**IL-15**	Monocytes	NK cells	Secretion of IFN-γ, macrophage colony-stimulating factor, and TNF-α; Work minimal synergism with IL-12.	[186]
**IL-22**	αβ T-cell classes Th1, Th22, and Th17, along with γδ T cells, NKTs, ILC3, neutrophils, and macrophages	HSCs	Reduces fat accumulation and steatosis; Induces senescence in HSCs.	[31,187,188,189]
**IL-30**	Th2 cells upon activation	NKT and HSCs	Attenuates liver fibrosis through inducing NKG2D–rae1 interaction.	[190]

**Table 6 cells-11-01500-t006:** Studies of liver fibrosis using mesenchymal stem cells.

NCT No.	Sponsor	Target Diseases	Clinical Trial Design	Clinical Phase	Status	Number of Patients	References
NCT04243681	Asian Institute of Gastroenterology, India	Liver cirrhosis	Combination of autologous mesenchymal and hematopoietic stem cells infused in patients	Phase 4	No results reported	5 participants	[233]
NCT05080465	Ukraine Association of Biobank	Liver cirrhosis	Long-term follow-up autologous MSC therapy for patients with virus-related liver cirrhosis	Phase 3	Active	700 participants	[234]
NCT00976287	Sun Yat-Sen University	Liver fibrosis, chronic hepatitis C	Liver function was monitored by serum examination. The levels of serum alanine aminotransferase (ALT), total bilirubin (TB), prothrombin time (PT), and albumin (ALB) were examined at pre-transplantation, and 3 days to 2 years post-transplantation	Phase 2	Results not posted	50 participants	[235]
NCT01483248	Zhejiang University, China	Liver cirrhosis, fibrosis	Menstrual blood-derived stem cells can improve the disease conditions in patients with liver cirrhosis.	Phase 1/2	No results posted	50 participants	[236]
NCT01573923	Allian cells Bioscience Corporation Limited	Liver cirrhosis	Intravenous administration of umbilical MSCs for the treatment of patients with liver cirrhosis in the next three years.	Phase 1/2	No results reported	320 participants	[237]
NCT01877759	Chaitanya Hospital, India	Liver cirrhosis	Bone-marrow-derived autologous stem cells + human umbilical-cord-derived MSCs	Phase 1/2	No results reported	20 participants	[238]
NCT01342250	Shenzhen Beike Bio-Technology Co., Ltd.	Liver cirrhosis	Safety and efficacy of human umbilical cord (hUC)-MSC transplantation for patients with decompensated liver cirrhosis	Phase 1/2	No results reported	20 participants	[239]
NCT03254758	Rohto Pharmaceutical Co., Ltd., Japan	Decompensated liver cirrhosis	First-in-human study of ADR-001, adipose-derived mesenchymal stem cells (AD-MSCs)	Phase 1/2	Recruiting	27 participants	[240]
NCT00420134	Shahid Beheshti University of Medical Sciences, Iran	Liver failure, cirrhosis	Investigators try to separate MSCs from end-stage liver disease, and then these cells are differentiated into the progenitors of hepatocytes; finally, the investigators inject these cells into the portal vein under ultrasound guidance.	Phase 1/2	No results reported	30 Participants	[241]
NCT01454336	Royan Institute	Liver fibrosis	Pioglitazone and autologous bone marrow MSC transplantation.	Phase 1	Completed	3 participants	[242]

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
