# Peer review of "Recent Advancements in Antifibrotic Therapies for Regression of Liver Fibrosis"

_cells, 2022, doi:10.3390/cells11091500_

Round 1
Reviewer 1 Report
This review article is not focused and present some aspects in large detail (e.g. the role of uPAR / suPAR as a biomarker), while other aspects (e.g. imaging biomarkers, microbiota, matrix biology) are negelected.
The presentation of immune mechanisms appears also unbalanced. Several studies, e.g. using single-cell RNA sequencing or spatial techniques, highlighted contributions from macrophages and other myeloid cells (e.g. dendritic cells) to disease progression. These aspects are not well presented, while NK cells are being described in detail. Other cells such as CXCR6+ auto-aggressive T cells, platelets or B cells are completely missing in the picture. The authors should avoid the M1/M2 paradigm for macrophages, as this is outdated in the context of fibrosis.
Table 1. FXR is primarily regarded as a target in hepatocytes, its role in HSC is still controversial. The main studies for FXR in phase III (1x fibrosis F3, 1x cirrhosis) use obeticholic acid, not tropifexor.
Table 4 is very incomplete, because major clinical trials for NASH-fibrosis are lacking (e.g. semaglutide, aramchol etc.). The authors should be aware that 4 out of 5 phase III clinical trials for NASH have been prematurely terminated due to inefficacy of the compounds in interim analyses. This needs to be discussed, when presenting ongoing clinical trials.
This article needs heavy language editing. For instance (page 16): “At the moment, NASH remains the most frequently prescribed disease indication for new antifibrotic agents [181].” This is clearly wrong, there are no antifibrotic agents approved that could be described. The authors probably mean that clinical trials testing new antifibrotic agents focus on NASH as an etiology.
Author Response
Reviewer-1 responses
We thank the reviewer for his/her critical review and appreciate their suggestions. As per their comments the following changes have been made:
Point 1: review article is not focused and present some aspects in large detail (e.g. the role of uPAR / suPAR as a biomarker), while other aspects (e.g. imaging biomarkers, microbiota, matrix biology) are neglected.
Response 1: Thank you for pointing this out. We have revised the biomarker section and included recent advancements in the non-invasive biomarkers. (Line no.291) uPAR/suPAR biomarker details are reduced to make it balanced. We included the recent -omic based biomarkers in a new section entitled “Evolving biomarker candidates in liver fibrosis” (Line no. 369). In this review we particularly focused on serum based non-invasive biomarkers and imaging biomarkers are beyond the scope of our current manuscript. However, as per your comments we dedicated a paragraph for the role of the gut microbiota as a biomarker for liver fibrosis. (Line no 545).
Point 2: The presentation of immune mechanisms appears also unbalanced. Several studies, e.g., using single-cell RNA sequencing or spatial techniques, highlighted contributions from macrophages and other myeloid cells (e.g., dendritic cells) to disease progression. These aspects are not well presented, while NK cells are being described in detail. Other cells such as CXCR6+ auto-aggressive T cells, platelets, or B cells are completely missing in the picture. The authors should avoid the M1/M2 paradigm for macrophages, as this is outdated in the context of fibrosis.
Response 2: We have revised section 1.1 “Hepatic scar and extracellular matrix leading to fibrosis” and added recent, single-cell RNA sequencing-based studies highlighting the contribution of scar-associated macrophages in the contribution to liver fibrosis (Line no.176). Role of CXCR6+ auto-aggressive T cells, platelets has been added (Line no. 164) T helper 17, regulatory T cells, mucosa-associated invariant T cells (Line no. 148) and platelets is also explained (Line no. 219). We have upgraded the M1/M2 paradigm for macrophages according to recent studies and further included the subtypes of M2 macrophages (Figure no. 4).
Point 3: Table 1. FXR is primarily regarded as a target in hepatocytes, its role in HSC is still controversial. The main studies for FXR in phase III (1x fibrosis F3, 1x cirrhosis) use obeticholic acid, not tropifexor.
Response 3: We have fixed the error. The table sections have been changed, and an entire section on FXR and the action of Obeticholic acid has been added in Table 2.
Point 4: Table 4 is very incomplete, because major clinical trials for NASH-fibrosis are lacking (e.g., semaglutide, aramchol etc.). The authors should be aware that 4 out of 5 phase III clinical trials for NASH have been prematurely terminated due to inefficacy of the compounds in interim analyses. This needs to be discussed, when presenting ongoing clinical trials.
Response 4: As mentioned by the reviewer, Table 4 has now been modified in Tables 2 and 3 to better go with the flow of the paper and an entire section regarding the discussion based on clinical trials has been added from section 2.2 onwards.
Point 5: This article needs heavy language editing. For instance (page 16): “At the moment, NASH remains the most frequently prescribed disease indication for new antifibrotic agents [181].” This is clearly wrong, there are no antifibrotic agents approved that could be described. The authors probably mean that clinical trials testing new antifibrotic agents focus on NASH as an etiology.
Response 5: We agree with the reviewer, the above statement has been changed according to the reviewer’s suggestion. “The clinical trials testing new antifibrotic agents focus on NASH as an etiology” was written as opposed to the previous statement. Regarding the language editing, the majority of the paper has been heavily modified as per the various comments from the reviewers and hopefully solves issues related to editing.
Reviewer 2 Report
Comments to stimulate improvement::
This is an excellent, well-written, high quality review.
It is focussed upon fibrosis -targeted therapies for liver fibrosis.
However, a few portions diverge from this theme. It would be better to totally focus on fibrosis -targeted therapies, rather than diverge into other targets such as metabolic and inflammatory and others, such as occurs in Table 4.
Abstract: Could also include autoimmune hepatitis and that there are many genetic causes of liver fibrosis.
Fig 1. Why is TNF shown but TGF not shown?
Page 4 line 123, “the hepatic cell population”: is, instead, “the hepatic stellate cell population” intended here?
page 4 line 140: NK cells should be mentioned in this paragraph and Fig 1, because the importance/role of NK is mentioned elsewhere, in section 2.5.
Fig 2: Important figure. Please increase font sizes as much as is feasible, especially of labels on the diagrammatic graph.
Page 6 line 212: Why is PPAR written here; its target is metabolic, not fibrotic. Also, insertion of PPAR here has no context around it. If this review had included metabolic targets, it would get too large. Thus, it is wise to stay within the boundaries that these authors have set. An alternative is to review not just PPAR, but also all, not just a few, other potential NASH therapy targets such as FGF21, GLP1, ACSL1, etc. It is sensible that many molecules are excluded from this review due to being out of scope.
Fibrosis targets at early or preclinical stages of research, could be included, such as PDGF and CCN2 . CCN2 is briefly mentioned in two places; perhaps have a paragraph dedicated to CCN2?
Page 8 Fewer mf (activated myofibroblasts) is one cause of lower the levels of pro-fibrotic response but some studies show lowered functional output of mf without changing the numbers of mf. Also, consider the concept that mf activation can be partial, that is, lying on a continuum between full activation and quiescence. Cells need not be binary regarding ‘activation’ or a particular function.
Fig 4: fonts / fig could be larger. The fig and its legend do not mesh well: For example, it is not clear in the fig that TSG-6 pushes towards M2.
Section 2.5 heading needs to be changed to more accurately describe the paragraph it heads.
Fig 5: It is surprising that HCC is featured in any figure in this document, when the review topic is fibrosis. Perhaps de-emphasise HCC.
Fig 5: Please reconsider this fig. Some arrows appear identical but seem to have different meanings. NK cells are central to the fig, so should be at the centre. The legend needs references for statemnts made.
Table 4 is puzzling, because it introduces for the first time, at the end of the article, molecular targets in NASH research that are not addressed anywhere else in this review. Table 4 appears to be an assortment of clinical trials that include ACE inhibitors, statins, DPP4 and PPAR targeting (pioglitazone and evogliptin), AT1R , etc. This doesn’t fit and lacks context in the text.
Author Response
Reviewer-2 responses
We thank the reviewer for his/her critical review and appreciate their suggestions. As per their comments the following changes have been made:
Point-1. Abstract: Could also include autoimmune hepatitis and that there are many genetic causes of liver fibrosis.
Response 1. Has been added in the Abstract, line number 13.
Point-2. Why is TNF shown but TGF not shown?
Response 2. Tumor necrosis factor-α (TNFα) is an inflammatory cytokine involved in liver inflammation and sustained liver inflammation leads to liver fibrosis but in the case of TGF-β signaling participates in different stages of disease progression, from initial liver injury to fibrosis is another different aspect of causing fibrosis, so in this present diagrammatic representation we only show TNFα role in cursing fibrosis.
Point-3. Page 4 line 123. The hepatic cell population”: is, instead, “the hepatic stellate cell population” intended here?
Response 3. Error is fixed. We have added, “the HSC population”
Point 4. Page 4 line 140: NK cells should be mentioned in this paragraph and Fig 1, because the importance/role of NK is mentioned elsewhere, in section 2.5.
Response 4. NKs have been discussed with other immune cells from line 147 onwards and have been added in Figure 1 as well.
Point-5. Fig 2: Important figure. Please increase font sizes as much as is feasible, especially of labels on the diagrammatic graph
Response 5. Thank you for pointing this out. The font sizes have been increased.
Point-6. Why is PPAR written here; its target is metabolic, not fibrotic. Also, insertion of PPAR here has no context around it. If this review had included metabolic targets, it would get too large. Thus, it is wise to stay within the boundaries that these authors have set. An alternative is to review not just PPAR, but also all, not just a few, other potential NASH therapy targets such as FGF21, GLP1, ACSL1, etc. It is sensible that many molecules are excluded from this review due to being out of scope.
Response 6. Considering there were issues regarding Table 4 as well as reviewers' concerns about metabolic target inclusion or not. For ease of understanding, a section on some metabolic targets has been added in Section 2.2. and PPAR has been mentioned in that section. This should avoid some concerns about the inclusion of PPAR and help solve issues concerning clinical trials associated with Table 4 and the flow of the paper.
Point-7. Fibrosis targets at early or preclinical stages of research, could be included, such as PDGF and CCN2 . CCN2 is briefly mentioned in two places; perhaps have a paragraph dedicated to CCN2?
Response 7. A paragraph for the roles of both PDGF and CCN2 has been added in the paper in section 2.4: the “Role of growth factors in liver fibrosis”.
Point-8. Page 8 Fewer mf (activated myofibroblasts) is one cause of lower the levels of pro-fibrotic response but some studies show lowered functional output of mf without changing the numbers of mf. Also, consider the concept that mf activation can be partial, that is, lying on a continuum between full activation and quiescence. Cells need not be binary regarding ‘activation’ or a particular function.
Response 8. We agree that fibrosis can be affected by function as well as the number of myofibroblast and that mf can be partially activated. We have updated section by referring recent relevant studies (Line no 1065)
Point 9. fonts / fig could be larger. The fig and its legend do not mesh well: For example, it is not clear in the fig that TSG-6 pushes towards M2.
Response 9. The fonts on the figure and some changes in the figure have been edited and the legend revised accordingly.
Point 10. Section 2.5 heading needs to be changed to more accurately describe the paragraph it heads.
Response 10. Section 2.5 has been changed to section 2.6 in the revised draft and the name changed to the “Immune-mediated role of NK cells”.
Point 11. Fig 5: It is surprising that HCC is featured in any figure in this document, when the review topic is fibrosis. Perhaps de-emphasise HCC
Response 11. HCC has been removed from the figure and edited accordingly to focus on fibrosis.
Point 12. Fig 5: Please reconsider this fig. Some arrows appear identical but seem to have different meanings. NK cells are central to the fig, so should be at the centre. The legend needs references for statemnts made.
Response 12. The arrows have been made identical and the NK cells have been placed in the center of the figure as suggested.
Point 13. Table 4 is puzzling, because it introduces for the first time, at the end of the article, molecular targets in NASH research that are not addressed anywhere else in this review. Table 4 appears to be an assortment of clinical trials that include ACE inhibitors, statins, DPP4 and PPAR targeting (pioglitazone and evogliptin), AT1R , etc. This doesn’t fit and lacks context in the text.
Response 13. As was mentioned earlier, the clinical trials table is considered necessary for this review with a majority of current clinical trials focusing on molecular targets as a means for fibrosis resolution. It was deemed important to briefly explain a few molecular targets in this review. Due to this reason a new section 2.2 “The role of Metabolic agents” was added and table 4 was broken down into 2 parts, namely tables 2 and 3 based on the target disease (NASH/NAFLD or directly liver fibrosis) as well as the mechanism of action. The idea of the 2 tables was simply to keep them short, clear, and precise.
Reviewer 3 Report
This manuscript focus on the antifibrotic therapies for chronic liver disease. The manuscript is easier to follow but it requires the following minor corrections:
Line 291_hMSCs is introduced in section 2.8, introduce in line 291.
Figure 2_introduce ACLF concept in the main text.
Figure 4_introduce STAT3 in the main text.
Table 4_add a column with the references of each study.
Authors should also refer to and discuss important studies on the development of portal hypertension mention in line 52.
Abbreviations lists is missing.
Author Response
Reviewer -3 Comments.
We thank the reviewer for his/her critical review and appreciate their suggestions. As per their comments the following changes have been made:
Point 1. Line 291_hMSCs is introduced in section 2.8, introduce in line 291.
Response 1: hMSCs have now been introduced in order of the first appearance in line 866 as suggested by the reviewer.
Point 2. Figure 2_introduce ACLF concept in the main text.
Response 2. To keep things easy to understand, ACLF has been changed to “Acute liver failure” studies which are referenced in (Line no. 417)
Point 3. Figure 4_introduce STAT3 in the main text.
Response 3. As suggested STAT 3 has been explained in the main text from line no 870 onwards.
Point 4. Table 4_add a column with the references of each study.
Response 4. Table 4 has been broken down into two tables, namely table 2 and table 3, and has been appropriately referenced as per suggestions.
Point 5. Authors should also refer to and discuss important studies on the development of portal hypertension mentioned in line 52.
Response 5. As suggested by the reviewer important studies regarding portal hypertension are mentioned from line 85 onwards.
Point 6. Abbreviations lists in missing.
Response 6. An abbreviations list is added at the end of the review file.
Reviewer 4 Report
This paper aims to present the current knowledge and understanding of the antifibrotic therapies for regression of liver. As well as explaining the mechanism involved in the development of liver fibrosis, the paper describes different therapeutic approaches that have been tested for the reversion of the process that includes the use of growth factors, cytokines, miRNAs, immune-based therapies, stem cell-based, and other approaches that target the extracellular matrix. Authors also presents some possible non-invasive biomarkers for the early detection of the disease.
It is well written, easy to read, and it clearly presents the subject. The bibliography include a number of recent papers in the field. The paper covers the topic from a general point of view, and could be for the interest of readers from the field of liver diseases, and more precisely liver fibrosis. In my opinion, this work is acceptable with some minor changes.
Minor points.
- Figure 2. Authors propose suPAR as serum biomarker. In the figure, there is a qualitative increase in the levels of this proposed biomarker. As mentioned in the footnote, “In patients with liver diseases, circulating suPAR levels increase with the increase in disease severity and are indicative of adverse prognosis”. Readers would appreciate to know whether this marker has been validated. Is there any quantitative correlation between some precise levels of this marker and phenotype? Any specific value/range to discriminate between patients that are Healthy or suffer from Chronic hepatitis, Fibrosis, Cirrhosis, Advance Cirrhosis?.
- Readers would also appreciate to know if the serum levels of this marker could be increased due to other causes not related to liver injury. Is it specific?.
- There are several mistakes in Figure 6.
- A) The figure caption explains that “Extracellular vesicles (EVs) derived from adipose tissue macrophage containing miR-155, 214, and 223 are taken up by liver cells, leading to increased insulin resistance by suppressing PPARG.”
In the figure, miR-155 derived from “adipose cells”, but miR-214 and miR-223 derived from HSC and neutrophils, respectively. Please, check the cellular source of those miRs.
- B) The figure caption explains that “miR-155, are taken up by liver cells, leading to increased insulin resistance by suppressing PPARG”.
In the figure, the “suppressing effects of the rest of miRs” have been expressed with a line and a perpendicular line.
- C) In the figure, the miRs have been coloured in red. miR-15b and miR-16b are represented in black.
- D) Which are the cellular source of miR-29a and miR-16?.
- E) In the figure, “endothelial inflamation” should be “endothelial inflammation”
- F) The figure caption explains that miR-195 downregulates cyclin E1, but in the Figure is miR-192
- G) The figure caption explains that miR-29b targets AKT, but in the Figure is miR-29a
- Apparently, there has been a problem with some symbols all along the manuscript. Most of them are missing (TGFβ, INFγ, NF-κB, α-SMA, PPARγ).
There have been these types of issues in Lines 49, 212, 248, 323, 332, 369, 393, 415, 459, 474, 484, 489.
- Table 1. The bibliography employed in section 2.2 is confusing for readers. In fact, the bibliography employed in Table 1 (references 74 to 82) is not mentioned in the text. For example, in Page 6 Line 218 when authors mention the results obtained by the mAb Lerdelimumab and Metelimumab, they refer to reference 70 and 71 in the text, but in table 1, the papers that are mentioned using the mAb Lerdelimumab and Metelimumab are references 74 and 75. When in Page 7 Line 223 when authors mention the results obtained inhibiting TIMP-1, they refer to reference 72 in the text, but in table 1, the papers that are mentioned using the target TIMP1 the references are 74 (again) and 76. When in Page 7 Line 228 when authors mention the results of the novel target integrin v6, they refer to reference 83 in the text, but in table 1, the paper that are mentioned using the target integrin v6 the reference is the 77.
- Abbreviations:
- Page 2 Line 64. The abbreviation (ECM) should be placed at the beginning of the line, just after “extracellular matrix”.
- The abbreviation of Hepatic stellate cells (HSC) is described in line 98. It can be used in Page 4 Line 127, in Page 11 Line 355, and in Page 12 Line 416.
- The abbreviations PPAR FXR (Page 6 lines 212) should be previously described.
- The abbreviation CTGF does not correspond to “collagen-derived growth factor” as metioned in (Page 8 line 254), but to “connective tissue growth factor” (also mentioned in the text).
- The abbreviation of natural killer (NK) has been described in line 275. It can be used in Page 10 Line 326, and in Page 11 Line 362.
- Please, include a reference after the following statements:
- Page 2 Line 70-71. “the non-fibrogenic type IV collagen is gradually replaced by the fibrogenic types I and II collagen.”
- Page 5 Line 171-172. “The importance of the coagulation cascade in the development of liver fibrosis has been established”.
- Page 12 Lines 390-391: ¨miR-29 is one of the most extensively studied and referenced microRNAs in the development of liver fibrosis”
- Page 12 Lines 404-405: “it has been demonstrated that the miR-15 family promotes cell proliferation and induces apoptosis”
- Figure 1: Please, include in the footnote the meaning of the “yellow component” closed to the “WPB”. Are the Weibel-Palade bodies?
- As mentioned in the text, hepatic stellate cells are one of the more relevant cell types involved in the development of liver fibrosis. In Page 4 line 100, authors mention the four distinct phenotypes (quiescent, activated, inactivated, and senescent) of these cells. Readers would appreciate a table summarizing the principal characteristics of each phenotype.
- Page 4 line 108-9. “In certain animal experiments, removing the causative agent can result in cirrhosis regression”. What about the regression of cirrhosis in human patients? Something is mentioned about patients suffering from viral hepatitis, but a more general view of this topic would be appreciated.
- Page 5 Line 154. The word “fibro lysis” has been employed together as “fibrolysis” in lines 113,122 and 208.
- Page 5 Lines 188-189. “Fibrosis severity in NAFLD and thus may be a promising biomarker candidate, as illustrated in Figure 2”. Please consider to rephrase this sentence.
- Figure 4. “The role of TSG-6 in cellular growth and proliferation in fibrotic liver cells”. Readers would appreciate that authors would concrete the cell types that are directly affected by the glycoprotein.
Author Response
Reviewer 4- Comments
We thank the reviewer for his/her critical review and appreciate their suggestions. As per their comments the following changes have been made:
Point 1: suPAR as serum biomarker
- Readers would appreciate to know whether this marker has been validated.
- Is there any quantitative correlation between some precise levels of this marker and phenotype? Any specific value/range to discriminate between patients that are Healthy or suffer from Chronic hepatitis, Fibrosis, Cirrhosis, Advance Cirrhosis?.
- Readers would also appreciate to know if the serum levels of this marker could be increased due to other causes not related to liver injury. Is it specific?.
Response 1:
- Although no specific study mentions whether suPAR has been validated, it mentioned that in clinical practice, a biomarker with a negative predictive value (NPV) of at least 90% can assist in the early identification of chronic liver disease. Scientific studies show that suPAR has a strong NPV and in the presence of chronic liver disease suPAR levels are elevated. Hence it was deemed important in this study as a non-invasive biomarker.
- Briefly, A study of 104 patients shows that the suPAR level on admission was superior to CRP, hematocrit, and creatinine as a prognostic marker of the severity of acute alcoholic pancreatitis. Using a cut-off value of 5 ng/ml, the sensitivity and specificity for predicting moderate or severe pancreatitis were 79% and 78%, respectively. The suPAR level was significantly associated with severity: Mild and moderate/serious: 4.2 ng/mL vs 6.2 ng/mL, respectively. Further details and references are mentioned in the paper from line no. 391 respectively.
- Since suPAR is an inflammation marker, it can be increased due to factors/diseases causing inflammation. Hence this form of prognostic biomarkers is purely used as an initial indicator of disease and further specific testing would be required to better understand the underlying cause of the inflammation. But according to a company that prepares test kits for suPAR detection called suPARnostic by ViroGates. It was useful in the detection of suPAR in cirrhosis, alcohol liver disease, excessive alcohol consumption, NAFLD, liver failure, HCC as well as alcoholic pancreatitis.
Point 2: Mistakes in Figure 6
- The figure caption explains that “Extracellular vesicles (EVs) derived from adipose tissue macrophage containing miR-155, 214, and 223 are taken up by liver cells, leading to increased insulin resistance by suppressing PPARG.” In the figure, miR-155 derived from “adipose cells”, but miR-214 and miR-223 derived from HSC and neutrophils, respectively. Please, check the cellular source of those miRs.
- The figure caption explains that “miR-155, are taken up by liver cells, leading to increased insulin resistance by suppressing PPARG”. In the figure, the “suppressing effects of the rest of miRs” have been expressed with a line and a perpendicular line.
- In the figure, the miRs have been coloured in red. miR-15b and miR-16b are represented in black.
- Which are the cellular source of miR-29a and miR-16?
- In the figure, “endothelial inflammation” should be “endothelial inflammation”
- The figure caption explains that miR-195 downregulates cyclin E1, but in the Figure is miR-192
- The figure caption explains that miR-29b targets AKT, but in the Figure is miR-29a
Response 2
- There was a slight error in writing this sentence as it was unclear in the legends. As is shown in Figure 6 and now rectified in the legends. miR155 is uptaken by adipose cells, mir214 by HSCs and miR223 by neutrophils, references of which are mentioned in line 1057.
- The suppression effects of miR155 via PPARϒ are now clearly shown in the figure.
- The color corrections have been made in the figure as suggested.
- Both miR29a are expressed in human peripheral blood mononuclear cells. miR16 are involved in the post-translational regulation of gene expression in circulatory system.
- The corrections have been made in the figure file as suggested
- The figure legends as well as the figure itself have been appropriately modified.
- The figure caption has been corrected.
Point 3. Symbols appear missing
- There has been a problem with some symbols all along the manuscript. Most of them are missing (TGFβ, INFγ, NF-κB, α-SMA, PPARγ).
- There have been these types of issues in Lines 49, 212, 248, 323, 332, 369, 393, 415, 459, 474, 484, 489.
Response 3. The symbols that were missing in the paper have all been rectified. Thank you for your thorough assessment.
Point 4: Table 1 related issues
- The bibliography employed in section 2.2 is confusing for readers. In fact, the bibliography employed in Table 1 (references 74 to 82) is not mentioned in the text. For example, in Page 6 Line 218 when authors mention the results obtained by the mAb Lerdelimumab and Metelimumab, they refer to reference 70 and 71 in the text, but in table 1, the papers that are mentioned using the mAb Lerdelimumab and Metelimumab are references 74 and 75.
- When in Page 7 Line 223 when authors mention the results obtained inhibiting TIMP-1, they refer to reference 72 in the text, but in table 1, the papers that are mentioned using the target TIMP1 the references are 74 (again) and 76.
- When in Page 7 Line 228 when authors mention the results of the novel target integrin v6, they refer to reference 83 in the text, but in table 1, the paper that are mentioned using the target integrin v6 the reference is the 77
Response 4:
a), b), c) All the appropriate changes regarding references and its flow in the main text as well as the table have now been rectified. The table is now numbered Table 4 and is in section 2.3
Point 5: Abbreviations
- Page 2 Line 64. The abbreviation (ECM) should be placed at the beginning of the line, just after “extracellular matrix”.
- The abbreviation of Hepatic stellate cells (HSC) is described in line 98. It can be used in Page 4 Line 127, in Page 11 Line 355, and in Page 12 Line 416.
- The abbreviations PPAR FXR (Page 6 lines 212) should be previously described.
- The abbreviation CTGF does not correspond to “collagen-derived growth factor” as metioned in (Page 8 line 254), but to “connective tissue growth factor” (also mentioned in the text).
- The abbreviation of natural killer (NK) has been described in line 275. It can be used in Page 10 Line 326, and in Page 11 Line 362.
Response 5: All rectifications suggested have been managed in the paper, a further abbreviations list has also been added in the end of the review paper.
Point 6: Missing references
- Page 2 Line 70-71. “the non-fibrogenic type IV collagen is gradually replaced by the fibrogenic types I and II collagen.”
- Page 5 Line 171-172. “The importance of the coagulation cascade in the development of liver fibrosis has been established”.
- Page 12 Lines 390-391: ¨miR-29 is one of the most extensively studied and referenced microRNAs in the development of liver fibrosis”
- Page 12 Lines 404-405: “it has been demonstrated that the miR-15 family promotes cell proliferation and induces apoptosis”
Response 6: All missing references have been sorted in the review file
Point 7: Please, include in the footnote the meaning of the “yellow component” closed to the “WPB”. Are the Weibel-Palade bodies?
Response 7: As is already mentioned by the reviewer, WPB is indeed Weibel-Palade bodies denoted by the yellow component and is mentioned in the legends now in brackets for further clarification.
Point 8: In Page 4 line 100, authors mention the four distinct phenotypes (quiescent, activated, inactivated, and senescent) of these cells. Readers would appreciate a table summarizing the principal characteristics of each phenotype.
Response 8: As was requested by the reviewer, a new table summarizing the characteristics of the HSC phenotypes namely table 1 has been created.
Point 9: Page 4 line 108-9. “In certain animal experiments, removing the causative agent can result in cirrhosis regression”. What about the regression of cirrhosis in human patients? Something is mentioned about patients suffering from viral hepatitis, but a more general view of this topic would be appreciated.
Response 9: References associated with fibrosis regression in human patients are added to further clarify the point as suggested by the reviewer. It’s marked in section 1.2.
Point 10 Minor edits
- Page 5 Lines 188-189. “Fibrosis severity in NAFLD and thus may be a promising biomarker candidate, as illustrated in Figure 2”. Please consider to rephrase this sentence.
- Page 5 Line 154. The word “fibro lysis” has been employed together as “fibrolysis” in lines 113,122 and 208.
- Figure 4. “The role of TSG-6 in cellular growth and proliferation in fibrotic liver cells”. Readers would appreciate that authors would concrete the cell types that are directly affected by the glycoprotein.
Response 10:
- The line has been removed and rephrased to fit better in the previous sentence.
- The word fibrolysis has now been changed to fibro lysis as suggested by the reviewer.
- As TSG-6 glycoprotein directly affects HSCs, so has been marked in the Figure 4 legends.
Round 2
Reviewer 1 Report
The work is very much improved.